

# Codominant grasses differ in gene expression under experimental climate extremes in native tallgrass prairie

Ava M. Hoffman[1,2], Meghan L. Avolio[3], Alan K. Knapp[1,2] and Melinda D. Smith[1,2]

[1] Department of Biology, Colorado State University, Fort Collins, CO, United States of America
[2] Graduate Degree Program in Ecology, Colorado State University, Fort Collins, CO, United States of America
[3] Department of Earth & Planetary Sciences, The Johns Hopkins University, Baltimore, MD, United States of America

Corresponding author
Ava M. Hoffman,
ava.hoffman@colostate.edu,
avamariehoffman@gmail.com

## ABSTRACT

Extremes in climate, such as heat waves and drought, are expected to become more frequent and intense with forecasted climate change. Plant species will almost certainly differ in their responses to these stressors. We experimentally imposed a heat wave and drought in the tallgrass prairie ecosystem near Manhattan, Kansas, USA to assess transcriptional responses of two ecologically important $C_4$ grass species, *Andropogon gerardii* and *Sorghastrum nutans*. Based on previous research, we expected that *S. nutans* would regulate more genes, particularly those related to stress response, under high heat and drought. Across all treatments, *S. nutans* showed greater expression of negative regulatory and catabolism genes while *A. gerardii* upregulated cellular and protein metabolism. As predicted, *S. nutans* showed greater sensitivity to water stress, particularly with downregulation of non-coding RNAs and upregulation of water stress and catabolism genes. *A. gerardii* was less sensitive to drought, although *A. gerardii* tended to respond with upregulation in response to drought versus *S. nutans* which downregulated more genes under drier conditions. Surprisingly, *A. gerardii* only showed minimal gene expression response to increased temperature, while *S. nutans* showed no response. Gene functional annotation suggested that these two species may respond to stress via different mechanisms. Specifically, *A. gerardii* tends to maintain molecular function while *S. nutans* prioritizes avoidance. *Sorghastrum nutans* may strategize abscisic acid response and catabolism to respond rapidly to stress. These results have important implications for success of these two important grass species under a more variable and extreme climate forecast for the future.

## INTRODUCTION

Climatic extremes, such as drought and heat waves, are predicted to increase in frequency and magnitude with forecasted climate change (*Stocker et al., 2013*). These extreme events may significantly impact ecosystem structure and function depending on their severity (*Ciais et al., 2005*; *García-Herrera et al., 2010*; *Smith, 2011*; *Knapp et al., 2015*). Climate extremes may affect plants in species-specific ways, such as through timing

of bud development (*Bokhorst et al., 2008*), variation in tissue die-back (*Kreyling et al., 2008*), and differences in physiological performance (*Hoover, Knapp & Smith, 2014a*) or chemical composition (*AbdElgawad et al., 2014*). Several recent studies have found species to respond differentially to extreme events like drought and heat waves (*Beierkuhnlein et al., 2011*; *Hoover, Knapp & Smith, 2014a*; *Nardini et al., 2016*). However, the mechanisms that lead to differences in plant performance are not always clear (*McDowell et al., 2008*). Understanding gene regulation may help explain the mechanisms of plant response to novel stressful environments (*Leakey et al., 2009*; *Swarbreck et al., 2011*). Gene regulation may also be more sensitive to periods of extreme climate compared with physiological performance and growth traits that may have delayed response. Likewise, gene regulation may reveal variation which can affect fitness, selection, and adaptation to new environmental conditions (*Ouborg & Vriezen, 2007*; *Gibson, 2008*; *Avolio & Smith, 2013*; *Vázquez et al., 2015*). However, most molecular studies of plant responses to drought and heat stress are focused on model organisms with limited ecological relevance (*Leakey et al., 2009*).

Here, we present a comparison of gene regulation responses of two $C_4$ grass species, *Andropogon gerardii* and *Sorghastrum nutans*, to an experimentally induced heat wave and drought in the field. These two dominant grasses are native to the tallgrass prairie ecosystem of the Central US and play an important role in determining community and ecosystem structure and function (*Smith & Knapp, 2003*; *Whitham et al., 2006*; *Whitham et al., 2008*; *Koerner et al., 2014*). They are often assumed to be functionally similar (i.e., both closely related phylogenetically, rhizomatous, $C_4$ warm-season tallgrasses, *Weaver & Fitzpatrick, 1934*; *Benson & Hartnett, 2006*; *Estep et al., 2014*), and both are relatively resistant to stress (*Knapp, 1985*; *Swemmer, Knapp & Smith, 2006*; *Tucker, Craine & Nippert, 2011*). However, *A. gerardii* and *S. nutans* differ in physiological response and abundance under different temperature and water availability (*Silletti & Knapp, 2002*; *Silletti, Knapp & Blair, 2004*; *Swemmer, Knapp & Smith, 2006*; *Nippert et al., 2009*; *Hoover, Knapp & Smith, 2014b*; *Hoover, Knapp & Smith, 2014a*). At the level of gene regulation, *A. gerardii* has been shown to be more sensitive to thermal stress (*Travers et al., 2007*; *Travers et al., 2010*; but see *Smith, Hoffman & Avolio, 2016*) while *S. nutans* is more sensitive to moderate water stress (*Smith, Hoffman & Avolio, 2016*; *Hoffman & Smith, 2017*). Specifically, *S. nutans* was more responsive to both a year-round 2 °C increase in temperature and more variable precipitation patterns (and lower average soil water availability) than *A. gerardii* (*Smith, Hoffman & Avolio, 2016*). *Sorghastrum nutans* also showed greater plasticity for dealing with water stress at the gene level (*Hoffman & Smith, 2017*). To date, much of the research assessing sensitivity of these grasses to heat and water stress has focused on chronic, subtle changes in temperature (2 °C increase in temperature) and water availability (on average 14% reduction in soil moisture; *Fay et al., 2011*). It remains unknown whether these two species would regulate genes differently under more extreme conditions, such as heat waves and droughts, which are predicted increase in frequency and severity in the Central US with climate change (*Cook, Ault & Smerdon, 2015*).

To increase our mechanistic understanding of the response of *A. gerardii* and *S. nutans* to climate extremes typical of the region, we analyzed the transcriptional profiles of both

grass species during an 18-day controlled heat wave under both watered and drought conditions within natural field plots. As in past research (*Travers et al., 2007*; *Travers et al., 2010*; *Smith, Hoffman & Avolio, 2016*), we measured gene expression using heterologous hybridization with cDNA microarrays designed for a closely related model species, *Zea mays*. We coupled the microarray data with filtering through each species' RNA-seq transcriptome (*Hoffman & Smith, 2017*). We hypothesized that gene regulation (number of genes, functional groups) would differ between *A. gerardii* and *S. nutans* in response to the heat wave under both watered and drought conditions, with these grasses employing different strategies for coping with extreme heat and water stress.

## METHODS

### Site description and experimental treatments

The study was carried out within the context of an existing long-term climate change experiment, the Rainfall Manipulation Plots (RaMPs), located at the Konza Prairie Biological Station in north-eastern Kansas (39°05′N, 96°35′W). Kansas State University, Manhattan, KS, USA and the Konza Prairie Biological Station granted explicit permission to the authors to sample with minimal impact within the RaMPs. The RaMPs is located in a native, annually burned site and consists of twelve 14 × 9 m greenhouse shelters (without walls) equipped with a clear (UV transparent) polyethylene roof to exclude natural rainfall inputs (*Fay et al., 2011*). Our experimental plots were located in two RaMPs (RaMP 12 and 13) in areas outside the 6 × 6 m experimental plots, but still located underneath the shelter infrastructure. Each of these areas is approximately 3 × 8 m in size, within which we located a 3 × 6 m experimental sampling plot. The RaMP 12 sampling plot was watered from late-May to mid-Aug to create a watered condition, whereas all ambient rainfall was excluded from the RaMP 13 sampling plot to create a drought. For both the watered and drought plots, a controlled high heat treatment was achieved by installing pairs of rectangular infrared heating lamps (Kalglo 2000 W; Kalglo Electronic Co Inc., Bethlehem, PA, USA) (Fig. S1). This resulted in a high heat treatment zone with a daytime target maximum of +8 °C above ambient midday temperature (Fig. S2), alongside ambient temperature treatment zones. The four treatments allowed us to examine the effects of drought and heat individually along with their interaction. The high heat treatment was imposed for an 18-day period (July 17 to August 4), when heat waves have generally occurred in the past (*Hoover, Knapp & Smith, 2014b*).

Prior to initiation of the experiment, canopy temperature in the watered sampling plot was measured using an infrared thermometer mounted on a movable platform (approx. 0.5 m above the canopy). Soil moisture was monitored at a depth of 0–15 cm with 30-cm time-domain reflectometry probes (Model CS616, Campbell Scientific, Logan, Utah, USA) inserted at a 45° angle (see Supplemental Information).

### Plant sampling and measurements

The focal species, *A. gerardii* and *S. nutans,* are both are rhizomatous C$_4$ grasses that reproduce primarily vegetatively via belowground buds on rhizomes (*Brejda, Moser & Waller, 1989*; *Carter & VanderWeide, 2014*), which form dense intermixed stands, making
it virtually impossible to differentiate between clones in the field (*Avolio, Chang & Smith, 2011*). We sampled individuals of *A. gerardii* and *S. nutans* from native populations growing within the experimental treatment plots during two sampling campaigns conducted at Day 4 and Day 18 of the heat wave. Each sampling campaign was conducted between 11:00 and 15:00 CDT to allow for collection of leaf temperature and water status (see below).

During each sampling campaign, we sampled two, morphological similar individuals (tiller or ramets, with 3–5 fully expanded leaves) of each species within the high heat zone and ambient temperature zone in both the watered and drought sampling plots ($n = 2$ samples per species, four treatments, and two campaign dates, or $n = 16$ per species, $N = 32$ total samples). While a sample size of two per species and treatment combination is relatively small, we believe this sample size was appropriate given that our focus was on broadly detecting interspecific differences under the high heat and drought conditions. Although we did not control for plant genotype, we collected our samples within a limited sampling area ($10 \times 10$ cm) to minimize genotypic differences among samples. Leaf tissue was collected from individuals located within each treatment within a five-minute window. For each individual, the first or second fully expanded leaf was randomly selected for genomic analysis to ensure similar leaf age. The entire leaf was clipped and immediately flash-frozen and stored in liquid nitrogen until brought to the laboratory. Immediately after, we measured leaf temperature ($T_{leaf}$) and midday leaf water potential ($\Psi_{mid}$) on the remaining fully expanded leaf. $T_{leaf}$ was measured using a LI-6400 system (LiCOR, Inc., Lincoln, NE, USA). The whole leaf was then collected for determination of mid-day leaf water potential (LWP) using an Scholander-type pressure chamber (PMS Instruments, Inc., Corvallis, OR, USA).

## RNA preparation and microarray hybridization

Leaf tissue samples were stored in an −80 freezer prior to RNA extraction. Total RNA was extracted from the 32 leaf samples for both species using TRIzol reagent (Invitrogen, Carlsbad, CA, USA) (*McCarty, 1986*), and further purified with the RNeasy kit (Invitrogen, Carlsbad, CA, USA). RNA quantity was measured by a NanoDrop spectrophotometer (Nanodrop products, Thermo Scientific, Wilmington, DE, USA). The verification of RNA quality, preparation of cDNA, and the subsequent steps leading to hybridization and array scanning were performed by Biotechnology Resources of Keck facility at Yale University (http://keck.med.yale.edu/). We used maize spotted cDNA arrays (SAM 1.2, GEO platform GPL4521) produced by the Center for Plant Genomics at Iowa State University for hybridization. The arrays included 15,680 maize cDNA probes (14,118 informative) isolated from maize ear tissue.

## Quality control of heterologous hybridizations

In total, there were eight hybridizations for each species per sampling campaign (Table S1). Array image data were collected using GenePix software (Version 6; Axon, Downingtown, PA, USA). Prior to normalization across arrays, features with obvious abnormality and saturated signal were flagged and excluded from statistical analysis. Two steps were taken to minimize the probability of mishybridization and sequence divergence between the

focal species and the model species (*Leakey et al., 2009*). First, we used stringent criteria by excluding spots with signal to noise ratios less than 3 or larger than 10 to decrease the inclusion of cross-hybridization artefacts (*Verdnik, Handran & Pickett, 2002*). Second, the cDNA sequences of the maize microarray SAM1.2 (18,862 sequences) were aligned against the de novo RNA-seq transcriptome data sets of *A. gerardii* and *S. nutans* (*Hoffman & Smith, 2017*), previously generated using Trinity (version 2.1.1, *Haas et al., 2013*). We only included BLASTN (*Altschul et al., 1990*; *Altschul et al., 1997*) hits with an *e*-value cutoff of $1e^{-10}$ and alignment length larger than 150 base pairs from the *A. gerardii* and *S. nutans* transcript data sets. After these two steps, 7,964 and 6,035 probe sequences were included in the analysis, accounting for 61.4% and 56.6% of the maize SAM 1.2 array probes for *A. gerardii* and *S. nutans* respectively. A total of 5,109 features were common to both species. Because features were screened by both the intensity of hybridization signal and sequence similarity, the intensity values of the included features were reliable for further expression analysis. These same techniques have also been validated previously using quantitative real-time PCR (qPCR) (*Smith, Hoffman & Avolio, 2016*).

## Array data normalization and statistical analysis

An important source of systematic errors in two-color microarray experiments is the different properties of the dyes used to label the two samples (*Tseng et al., 2001*; *Yang et al., 2001*; *Yang et al., 2002*) and the hybridization variability from array to array. We used dye-swap design for the same pair of samples in the hybridizations (Table S1) to account for the dye effect (*Dabney & Storey, 2007*). Background signals were removed from median signal intensity and modelled similarly to *Travers et al. (2010)* to remove the array and dye effect:

$$y_{ijk} = A_i + D_j + A_i D_j + \varepsilon_{ijk},$$

where $y$ is the median intensity for the $k$ th gene on each array ($i$) with each dye ($j$), $A$ is the array effect for each array ($i$), $D$ is the dye effect for each dye ($j$), $AD$ is the array $\times$ dye interaction, and $\varepsilon_{ijk}$ is the stochastic error. Residuals from this model were adjusted by the minimum value to produce all positive residuals. To examine overall statistical effects, we used the residuals in the following model:

$$r_{klmno} = S_l + W_m + T_n + C_o + S_l W_m + S_l T_n + W_m T_n + \varepsilon_{klmno},$$

where $r$ is the residual for each gene ($k$) with each species ($l$), water treatment ($m$), temperature ($n$), and sampling date ($o$), $S$ is the species effect, $W$ is the water treatment effect (plot), $T$ is the temperature effect, and $C$ is the sampling date effect. Residuals were used to generate $\log_2$ expression ratios for the four variables: species (*A. gerardii*/*S. nutans*), water treatment (watered/droughted), temperature (ambient/heated), date (day 4/day 18). Any genes with missing signals were removed. We plotted the $\log_2$ expression ratio against the $\log_{10}$ intensity for each gene and performed a loess correction to normalize each set of $\log_2$ values (Fig. S3). Then, for each gene without missing values, a linear model was performed to test each main effect (species, water treatment, temperature, and date) as well as selected interactions (species $\times$ water treatment, species $\times$ temperature, and water

PeerJ ______________________________________________

treatment × temperature). Because of the variation in genes present across arrays, each model was constructed only if appropriate data was present. In other words, to test species effect, both species had to express the given gene. *P*-values were adjusted using a Bonferroni correction to account false discovery across multiple tests. All analyses were performed using R (version 3.3.2).

### Functional annotation, enrichment, and clustering

The functional annotation of transcripts was based on the Trinotate pipeline (version 3.0.1). We matched microarray probe sequences to known sequences using BLAST against the SwissProt annotated database (*Apweiler et al., 2004*), identified protein sequence homology using HMMER and Pfam (*Finn, Clements & Eddy, 2011*; *Finn et al., 2015*), and searched for known annotations within eggNOG and GO databases (*The Gene Ontology Consortium, 2015*; *Huerta-Cepas et al., 2016*). Ontology enrichment was determined using GOSeq (version 3.4, *Young et al., 2010*), a statistical package for R which accounts for multiple testing as well as differing probe lengths. Finally, clustering of gene modules was performed using the WGCNA package for R (version 1.51, *Langfelder & Horvath, 2008*) with a minimum module size of five genes.

## RESULTS

### Efficacy of the heat wave and drought treatments and impacts on $T_{leaf}$ and $\Psi_{mid}$

On average, the heated (heat wave) treatment resulted in an 8 °C increase in canopy temperature (Fig. S2A). On average, the drought treatment decreased volumetric soil water content from 28% to 24% midway through the heat wave (day 9). The high heat treatment further decreased soil water content by 2% for the watered and 5% for the drought treatments (Fig. S2B). Overall, the combined effect of drought and heat resulted in a drop from 29% to 22% volumetric water content. The increase in canopy temperature with the high heat treatment was reflected in greater leaf temperature ($T_{leaf}$) for both species; *A. gerardii* and *S. nutans* had significantly higher $T_{leaf}$ at both day 4 and 18 of the heat wave (Fig. S4). water content with the drought and high heat treatment were reflected in greater water stress in both species (i.e., more negative $\Psi_{mid}$, Fig. S4). For *A. gerardii*, the high heat treatment caused a large decrease in $\Psi_{mid}$, with this decline greatest at day 4 of the heat wave combined with drought ($-0.9$ MPa, Fig. S4). The decrease in $\Psi_{mid}$ with the high heat treatment was most pronounced in *S. nutans* after 18 days of heat wave under drought ($-1.7$ MPa, Fig. S4).

### Environment affects gene regulation in *A. gerardii* and *S. nutans*

Overall, 1,131 genes were shared across both species, 1,515 were shared across water treatment, 1,653 were shared across temperature treatment, and 1,390 were shared across date. Species ($p < 0.001$), water treatment ($p < 0.001$), and their interaction ($p < 0.001$) most significantly impacted gene expression. In other words, species gene expression response strongly depended on the drought environment. Temperature was only a weakly significant predictor of gene expression ($p = 0.048$) with no significant species

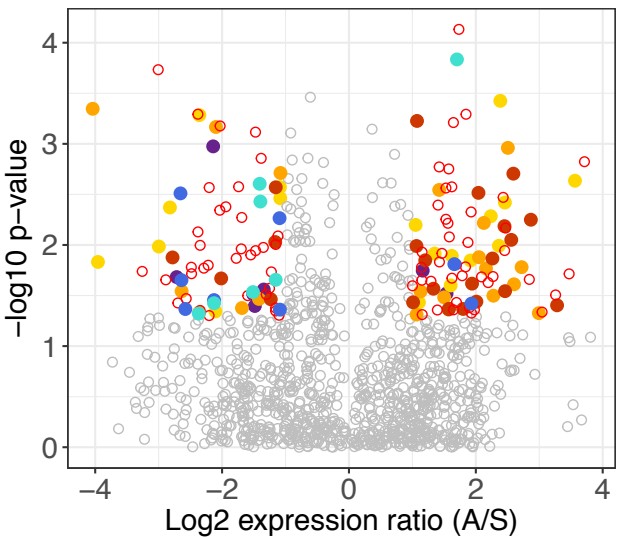

**Figure 1** **Differentially expressed genes in *A. gerardii* and *S. nutans.*** Differentially expressed genes have a log$_2$ fold change greater than one, shown as colored points where $p < 0.05$. Positive values indicate greater expression in *A. gerardii* while negative values indicate greater expression in *S. nutans*. Filled circles represent genes belonging to selected Gene Ontology groups. Open circles: red, differentially expressed, gray, not significantly different.

by temperature interaction. Gene expression did not vary across sample date/duration of the heat wave.

## Overall differences between *A. gerardii* and *S. nutans*

Of 1,131 genes found in both species, 160 differed significantly in their regulation between species. Genes with greater expression in *A. gerardii* were enriched in cellular metabolic process, biological regulation, and protein metabolic process, while genes with greater expression in *S. nutans* were enriched in negative regulation of metabolism, biological, and cellular processes, macromolecule catabolic process, and protein kinase activity (Fig. 1). Within cellular metabolic process, the most extreme differences were found in a methyltransferase and other transferases, GTP binding protein, Dihydrouridine synthase (Dus), as well as several transcription factors (Table 1). Among biological regulation genes, several transcription factors were strongly upregulated in *A. gerardii*. Protein metabolic processes included several ribosomal-related genes as well as fibrillarin upregulated in *A. gerardii*. Within genes significantly upregulated in *S. nutans*, the negative regulation (inhibition) category consisted of a finger protein as well as several membrane proteins like CMP-sialic acid transporter homolog (Table 1). Macromolecule catabolism genes included several proteasomes, 1,2-alpha-mannosidase, and a ubiquitin-conjugating enzyme. Among genes annotating to the term "stress", 18 were upregulated in *S. nutans* versus 31 upregulated in *A. gerardii*. Genes annotating broadly to "regulation" showed 91 upregulated in *A. gerardii* versus 74 in *S. nutans*.

Gene clustering was performed for day 18 samples to detect species differences for both plots at the end of the heat wave. Similarly regulated modules or groups of genes may lead

**Table 1** Selected differentially expressed genes.

| Maize gene | Description | Log₂ fold-change | Upregulated in | GO category |
|---|---|---|---|---|
| **Regulation between *A. gerardii* and *S. nutans*** | | | | |
| CB331760 | methyltransferase | 3.56 | *A. gerardii* | Cellular metabolic process |
| DV621283 | GTP binding protein | 3.28 | *A. gerardii* | Cellular metabolic process |
| DV490673 | Dihydrouridine synthase (Dus) | 2.99 | *A. gerardii* | Cellular metabolic process |
| DV491165 | transcription factor | 2.60 | *A. gerardii* | Biological regulation |
| BM331929 | transcription factor | 2.56 | *A. gerardii* | Biological regulation |
| CD510408 | fibrillarin | 2.56 | *A. gerardii* | Protein metabolic processes |
| DV491840 | finger protein | −2.66 | *S. nutans* | Negative regulation |
| DV491692 | CMP-sialic acid transporter homolog | −2.64 | *S. nutans* | Negative regulation |
| DV942581 | Proteasome | −2.37 | *S. nutans* | Macromolecule catabolism |
| DV490558 | 1,2-alpha-mannosidase | −2.13 | *S. nutans* | Macromolecule catabolism |
| DV493085 | ubiquitin-conjugating enzyme | −1.51 | *S. nutans* | Macromolecule catabolism |
| **Regulation within *A. gerardii*** | | | | |
| CB331250 | RNA-binding protein | −1.01 | *Drought* | Osmotic stress |
| CA989232 | ribosomal protein S3 | −1.51 | *Drought* | Osmotic stress |
| BM347878 | aconitate hydratase | −1.17 | *Drought* | Osmotic stress |
| CD651535 | histone acetyltransferase | −1.33 | *Drought* | Chromatin silencing |
| CB815849 | histone acetyltransferase | −1.86 | *Drought* | Chromatin silencing |
| DY576254 | Hsp70 protein | −1.06 | *Heat wave* | Protein folding |
| CD662140 | high mobility group-box domain | 1.08 | *Ambient temp.* | DNA binding |
| **Regulation within *S. nutans*** | | | | |
| DV489871 | ERBB-3 binding ribonuleoprotein | 1.33 | *Watered* | ncRNA metabolism |
| DV489639 | serrate RNA effector molecule | 1.24 | *Watered* | ncRNA metabolism |
| DV943322 | pseudouridine synthase | 1.16 | *Watered* | ncRNA metabolism |
| DV942798 | ribosome production factor 2 | 1.04 | *Watered* | ncRNA metabolism |
| BM073337 | polyribonucleotide nucleotidyltransferase | 1.02 | *Watered* | ncRNA metabolism |
| CD651136 | Cysteinyl-tRNA synthetase | 2.14 | *Watered* | ncRNA metabolism |
| BM078961 | methionine-tRNA ligase | 1.26 | *Watered* | ncRNA metabolism |
| CD651793 | valine-tRNA ligase with editing activity | 1.24 | *Watered* | ncRNA metabolism |
| DV492155 | aquaporin NIP3-1 | 1.22 | *Watered* | Transmembrane activity |
| BM340348 | NEP1-interacting protein | 1.19 | *Watered* | Methyltransferase activity |
| DV492743 | transcriptional corepressor | 1.17 | *Watered* | Negative regulation of transcription |
| CD527890 | E3 ubiquitin ligase SUD1 | −2.24 | *Drought* | Osmotic stress |
| DV489949 | aldo-keto reductase | −2.16 | *Drought* | Osmotic stress |
| BM348293 | hydrophobic protein LTI6A | −1.01 | *Drought* | Osmotic stress |
| DV491692 | CMP-sialic acid transporter homolog | −2.35 | *Drought* | Encapsulating structures |
| DV492287 | phosphatidylinositol kinase | −2.12 | *Drought* | Encapsulating structures |
| BM333861 | pectin acetylesterase 8 | −1.65 | *Drought* | Encapsulating structures |
| DV491662 | 26S protease | −2.40 | *Drought* | Catabolism |
| DV492129 | DNA-directed RNA polymerase II Rpb7p | −2.35 | *Drought* | Catabolism |
| DV492287 | phosphatidylinositol kinase | −2.12 | *Drought* | Catabolism |
| DV942393 | GDP-mannose 4,6 dehydratase | −4.99 | *Drought* | Organophosphate metab |
| DV493244 | triosephosphate isomerase | −2.47 | *Drought* | Organophosphate metab |
| DV491451 | phosphatidylinositol-4-phosphate 5-kinase | −2.10 | *Drought* | Organophosphate metab |

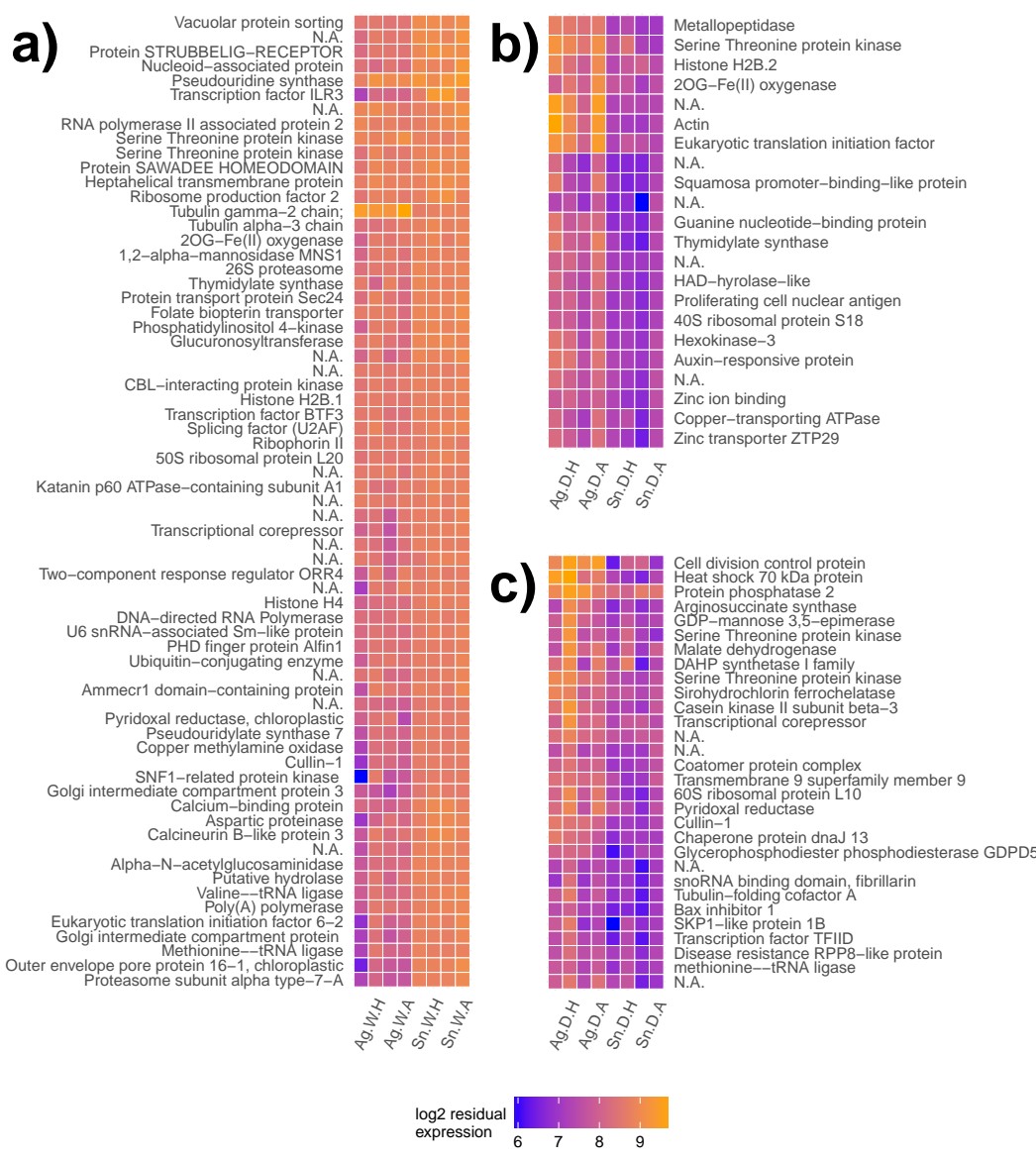

**Figure 2 Gene modules explaining species differences under different water availability.** Gene modules detected explaining species differences in watered (A) and drought (B, C) conditions. Sample names are presented on the *x*-axis, where each label applies to two columns of the same description (e.g., Ag. W.H applies to the first two columns, but both are replicates of *A. gerardii* in Watered plot with Heated treatment). Ag, *A. gerardii*, Sn, *S. nutans*, W, watered, D, drought, H, heated, A, ambient temperature. No annotation found, N.A.

to a greater understanding of gene networks contributing to different species responses. One gene module significantly explained species differences in the watered treatment ($p < 0.001$, Fig. 2A) with genes generally expressed more highly in *S. nutans*. Two gene modules significantly explained species differences in the drought treatment ($p = 0.01$, Fig. 2B and $p = 0.02$, Fig. 2C respectively). Under drought, genes generally had lower expression in *S. nutans*.

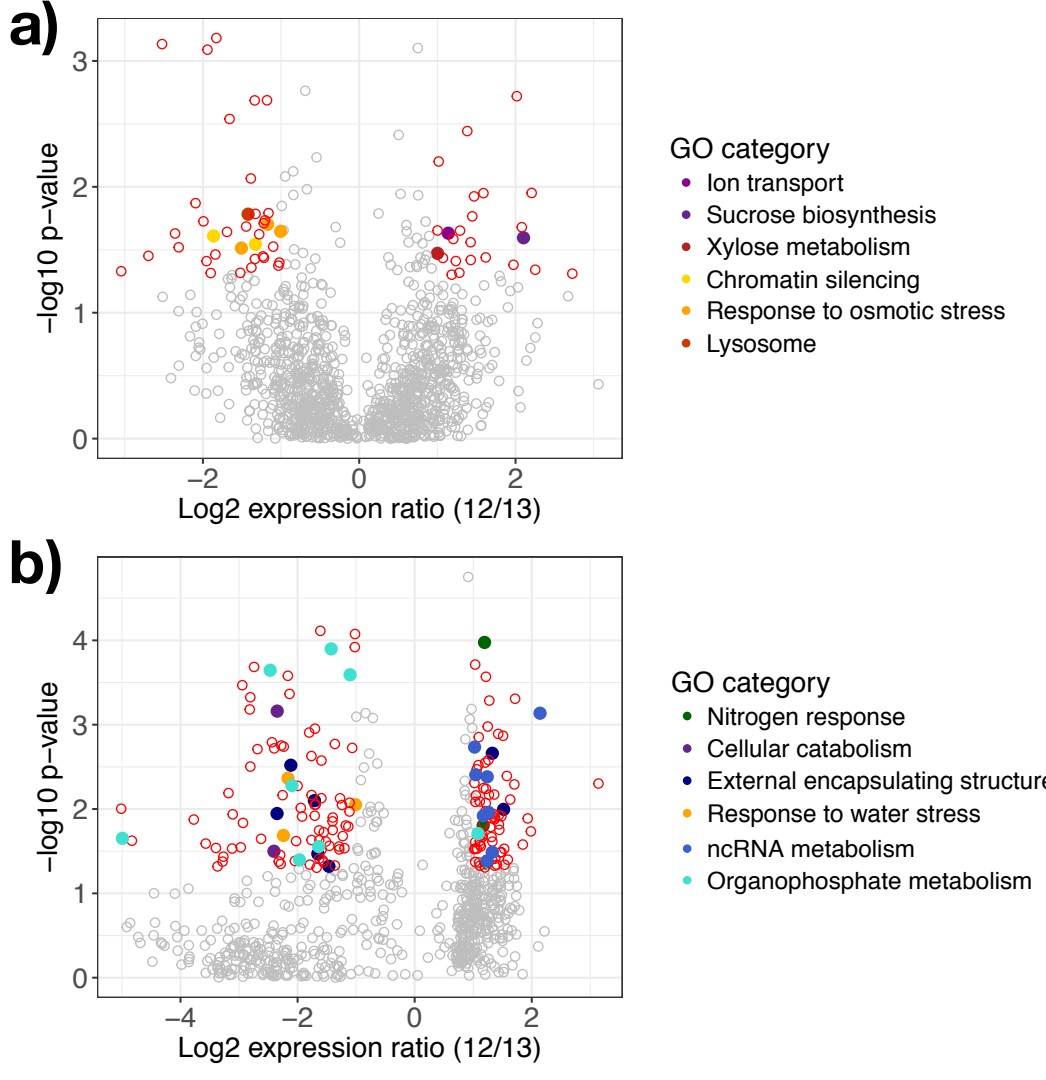

**Figure 3** **Differentially expressed genes in response to water availability.** Differentially expressed genes in (A) *A. gerardii* and (B) *S. nutans* only compared between watered and drought plots (12 and 13). Significantly different genes with $\log_2$ fold change greater than one are represented by colored points where $p < 0.05$. Positive values indicate greater expression in the watered plot while negative values indicate greater expression in the drought plot. Filled circles represent genes belonging to selected Gene Ontology groups. Open circles: red, differentially expressed, gray, not significantly different.

## Genes regulated in *A. gerardii*

In *A. gerardii*, 61 genes were significantly regulated in response to drought (5% of 1,148 total genes), with 24 genes upregulated under watered conditions and 37 upregulated under drought conditions. Few GO categories had strong enrichment (i.e., few genes per category). The drought treatment showed enrichment in response to osmotic stress, chromatin silencing, and lysosome. The watered treatment suggested greater abundance of xylose metabolism, sucrose metabolism, and ion transport (although each group contained only one gene) (Fig. 3A). Osmotic stress genes included an RNA-binding

protein, ribosomal protein S3, and aconitate hydratase (Table 1). Within chromatin silencing genes, two histone acetyltransferases were upregulated under drought. Among all genes, 24 genes annotating to "stress" were upregulated in the watered treatment, versus 29 under drought. Only two genes (both within *A. gerardii*) responded significantly to temperature. One gene was upregulated in response to higher temperatures (Hsp70 protein); another was downregulated under higher temperatures (high mobility group-box domain).

### Genes regulated in *S. nutans*

*Sorghastrum nutans* regulated more genes in response to drought than *A. gerardii* (23% of 762 genes total). Of these, 92 showed greater expression in the watered treatment while 82 showed greater expression under drought. Genes upregulated in the watered treatment showed GO enrichment in non-coding RNA (ncRNA) and RNA metabolism and nitrogen response. Genes upregulated under drought showed enrichment in response to water stress, external encapsulating structure, organophosphate metabolism, and cellular catabolism (Fig. 3B). Within the watered treatment ncRNA metabolism genes including ERBB-3 binding ribonuleoprotein, serrate RNA effector molecule, and pseudouridine synthase were upregulated (Table 1). *Sorghastrum nutans* in the watered treatment also showed greater expression of aquaporin NIP3-1, NEP1-interacting protein, and a transcriptional corepressor.

In contrast, *S. nutans* under drought showed greater expression of osmotic stress genes E3 ubiquitin ligase SUD1, 9 aldo-keto reductase, and hydrophobic protein LTI6A (Table 1). Among encapsulating structures, CMP-sialic acid transporter homolog, phosphatidylinositol kinase, pectin acetylesterase 8, and two glucuronosyltransferases (ranged from fold change of −1.47 to −1.71) were upregulated under drought. Catabolism genes within the drought treatment included 26S protease, DNA-directed RNA polymerase II Rpb7p, and phosphatidylinositol kinase. Lastly, the drought treatment showed increased expression of organophosphate metabolism genes including GDP-mannose 4,6 dehydratase, triosephosphate isomerase and phosphatidylinositol-4-phosphate 5-kinase. Among all genes, 12 (1.5%) genes annotating to "stress" were upregulated in the watered treatment, versus 20 (2.6%) under drought.

## DISCUSSION

Increasingly, ecological studies are using molecular techniques to study gene-level responses to global change in non-model organisms (*Travers et al., 2007*; *Leakey et al., 2009*; *Alvarez, Schrey & Richards, 2015*; *Smith, Hoffman & Avolio, 2016*). Genomic tools like microarrays have revealed mechanisms behind plant environmental responses in natural plant populations (*Jackson et al., 2002*; *Travers et al., 2007*; *Ungerer, Johnson & Herman, 2008*; *Leakey et al., 2009*; *Travers et al., 2010*; *Smith, Hoffman & Avolio, 2016*). Heterologous hybridization has proven useful for studying non-model organisms when the proper precautions are taken and stringent criteria are utilized to control for mishybridizations (*Leakey et al., 2009*; *Travers et al., 2010*; *Alvarez, Schrey & Richards, 2015*). Both environmental (*Gong et al., 2005*; *Hammond et al., 2006*; *Sharma et al., 2006*;
*Travers et al., 2010*; *Alvarez, Schrey & Richards, 2015*) and biotic (*Horvath & Llewellyn, 2007*; *Broz et al., 2008*) stress responses have been explored. Our study used heterologous hybridization to compare transcriptional responses of two non-model grasses under field conditions. We used stringent criteria to control for mishybridizations, multiple steps to normalize the array data, and sequence alignment with RNA-seq transcriptomes. One significant caveat of the microarray technique is the inability of microarray technology to distinguish between two scenarios: no microarray signal due to true low expression versus no microarray signal due to probe-to-gene mismatch. In other words, this study is limited to low versus high expression contrasts while excluding presence/absence analysis, and may fail to detect larger, significant shifts in gene expression. Moreover, these two species have almost certainly evolved unique genes to adapt to harsh conditions sometimes experienced in the tallgrass prairie. These species-specific genes may be the most insightful but are undetectable using these methods.

We expected that *A. gerardii* and *S. nutans*, two closely related and functionally similar species, would differ in their gene responses to heat-wave and drought. Specifically, *S. nutans* would regulate a greater number of genes from different groups compared with *A. gerardii*. This expectation was based on past evidence for greater sensitivity of *S. nutans* to more moderate water stress (*Nippert et al., 2009*; *Hoover, Knapp & Smith, 2014a*; *Smith, Hoffman & Avolio, 2016*). Overall, our hypothesis was supported; *S. nutans* had greater sensitivity to the imposed drought compared to *A. gerardii* in the percentage of regulated transcripts. Despite similar $T_{\text{leaf}}$ and $\Psi_{\text{mid}}$ measurements, *A. gerardii* appeared less responsive with a smaller proportion of genes (5%) exhibiting a significant change under drought. A similar pattern of gene regulation was observed when *A. gerardii* and *S. nutans* were exposed to more moderate changes in water availability in the field (*Smith, Hoffman & Avolio, 2016*). Thus, in line with past research, our results suggest that *A. gerardii* is more resistant to and/or better able to cope with water stress than *S. nutans*. According to gene modules detected using statistical clustering, *S. nutans* genes tended toward downregulation on day 18 of the drought compared to *A. gerardii* (Fig. 2), which could represent a surpassed stress response threshold. *Sorghastrum nutans* has also shown loss of function under stress with respect to net photosynthetic rate and biomass production (*Hoover, Knapp & Smith, 2014a*).

Despite strong support for the non-additive effects of water and temperature stress in some systems (*Atkinson & Urwin, 2012*; *Johnson et al., 2014*; *Suzuki et al., 2014*), the two did not show a significant interaction. However, previous work comparing these two species also found no environmental interaction (*Hoover, Knapp & Smith, 2014a*). In our study, only two genes within *A. gerardii* responded to the high heat treatment. Previous ecophysiological research has shown greater relative temperature sensitivity in *A. gerardii* (*Nippert et al., 2009*). Gene expression did not vary across sampling date, despite evidence for plasticity in other species (*Hayano-Kanashiro et al., 2009*; *Meyer et al., 2014*). However, it is important to acknowledge that fewer genes overlapped across sample date, and only these genes were contrasted. Many genes may have been expressed during the first sampling date but not during the second date and vice versa.

Over all treatments, *A. gerardii* tended to have greater expression of metabolic and regulatory genes compared to *S. nutans*, suggesting it maintains high levels of metabolic function in many environmental conditions and may strategize plasticity at the regulatory level (i.e., utilizes more transcription factors, tRNA enzymes, and ribosomal enzymes). In other words, gene expression remains fairly constant but may be modified downstream. Expression of transcription factors has been widely implicated in drought adaptation and response (*Yamaguchi-Shinozaki & Shinozaki, 2006*; *Yoshida et al., 2015*; *Kudo et al., 2016*; *Gahlaut et al., 2016*). On the other hand, greater transcription of negative regulators and catabolism genes in *S. nutans* may reflect an ability to respond more rapidly to drought stress. Over-expression of negatively regulating PHD finger proteins in *Arabidopsis* inhibits pathways and leads to enhanced stress tolerance (*Wei et al., 2015*) and the 26S proteastome helps modulate ABA response as well as degrade proteins not needed under non-stressed conditions (*Stone, 2014*). Both species appear equipped to handle stressful conditions, though *S. nutans* seems to focus on rapid response via molecular breakdown and pathway inhibition whereas *A. gerardii* maintains higher levels of metabolic process and regulates transcription via transcription factors. Due to multiple statistical tests performed, only the most significant genes responding to drought were examined. Only two of these overlapped in *A. gerardii* and *S. nutans*, further highlighting their different drought response strategies.

*Andropogon gerardii* has previously shown greater ecophysiological response to temperature (*Nippert et al., 2009*), but may actually be less sensitive at the gene expression level to mild temperature stress (*Smith, Hoffman & Avolio, 2016*). A consensus regarding temperature response may remain elusive considering only two genes significantly responded to temperature in *A. gerardii*. Hsp70 is well known to be upregulated under stress to assist protein folding (*Hayano-Kanashiro et al., 2009*; *Wang et al., 2015*), while high mobility group (HMG) genes are known to be negatively correlated with stress response (*Kim et al., 2010*). The general lack of response may be due to our stringent gene filtering criteria, but may also reflect presence of unique genes in these species. Non-targeted methods (such as RNA-seq, *Hoffman & Smith, 2017*) have been successful in these species and would likely reveal more comprehensive differences under temperature extremes.

Of osmotic stress-related genes upregulated in *A. gerardii* in response to drought, Glycine-rich RNA-binding protein 2 is known to have RNA chaperone activity during abiotic stress (*Kim et al., 2007*), 40S ribosomal protein may be upregulated to compensate for mild osmotic stress (*Ma et al., 2016*), and aconitate hydratase has been shown to increase under water and heat stress (*Johnson et al., 2014*) in a compensatory manner due to its sensitivity to oxidative damage (*Budak et al., 2013*). Osmotic stress-related genes were also upregulated in *S. nutans* under drought, however their function was quite different. E3 ubiquitin ligase is understood to play a role in regulating response to ABA (*Doblas et al., 2013*; *Zhao et al., 2014*), aldo-keto reductase 4C9 is involved in scavenging toxins produced under stress (*Simpson et al., 2009*), and hydrophobic LTI6A is a transmembrane protein which responds to low temperature stress, drought, and ABA (*Wang et al., 2016*). These focal genes tied to osmotic stress response suggest that while both species are responding to drought, their strategies differ. In this case, *S. nutans* not only regulates a greater percentage of genes but also focuses on ABA response, whereas *A. gerardii* appears to upregulate genes

to compensate for lost function. Among its many roles, ABA may help with stomatal closure and drought avoidance (*Jones & Mansfield, 1970*).

Within *S. nutans*, ncRNAs (transcriptional regulators) declined under drought, which have been shown to downregulate in response to drought (*Hackenberg et al., 2015*). In this study, many of these genes mapped to transcription factors or RNA binding, which are typically upregulated under drought (*Yamaguchi-Shinozaki & Shinozaki, 2006*; *Yoshida et al., 2015*; *Kudo et al., 2016*; *Gahlaut et al., 2016*; but see *Baldoni, Genga & Cominelli, 2015*). This could indicate that *S. nutans* experienced mechanistic loss of function under drought conditions. Catabolism related genes upregulated under drought may indicate salvaging of important functions. For example, phosphatidylinositol-4-phosphate 5-kinase is known to modulate ABA response as well as prevent breakdown of proline, an important ROS scavenger (*Leprince et al., 2014*). The 26S protease regulatory subunit lends additional breakdown of molecules potentially involved in signaling (*Stone, 2014*). Similarly, RNA polymerase subunit Rpb7p is thought to help degrade mRNAs as a counteractive measure (*Shalem et al., 2011*). Of the genes not involved in cellular catabolism, some were tied to cell wall integrity (e.g., pectin acetylesterase) and may serve as a last resort for survival under extreme stress (*Houston et al., 2016*). Meanwhile, few genes suggested loss of function or disassembly role in *A. gerardii*, which further emphasizes *S. nutans*' greater sensitivity to drought stress. Of note is *A. gerardii*'s more consistent regulation of stress transcripts: this species shifted from 2.1% to 2.5% "stress" annotations following drought, while *S. nutans* shifted from 1.6% to 2.6% "stress" annotations. This could mean that *A. gerardii* tolerates stress and avoids sensitivity by constitutively expressing some stress responses. This makes sense considering the broad array of stressors *A. gerardii* is likely to experience (*Hulbert, 1988*; *Turner & Knapp, 1996*; *Silletti, Knapp & Blair, 2004*; *Swemmer, Knapp & Smith, 2006*; *Koerner et al., 2014*). Overall, these results suggest that *S. nutans*' ecophysiological sensitivity may be mechanistically tied to downregulation of genes under stress coupled with rapid avoidance strategies, such as the regulation of ABA. *Andropogon gerardii*'s apparent lack of sensitivity may result from upregulation of stress sensitive transcripts coupled with maintenance of cellular processes despite extreme stress.

## CONCLUSIONS

Our results suggest that *A. gerardii* is more resistant to extremes in water stress and does not downregulate as many processes as *S. nutans*. Surprisingly, response to the heat wave was minimal. While *A. gerardii* contributes proportionally more aboveground biomass (*Smith & Knapp, 2003*) and is an important mediator of species diversity in the tallgrass prairie ecosystem (*Collins, 2000*; *Smith et al., 2004*), *S. nutans* is able to attain greater photosynthetic rates that could be linked to carbon storage (*Hoover, Knapp & Smith, 2014a*). Differences in sensitivity and stress response mechanisms could ultimately alter community structure and ecosystem function in the tallgrass prairie ecosystem.

## ACKNOWLEDGEMENTS

We thank S Yuan for preparing the samples for this study. JM Blair in part conceived the rainfall manipulation plots experiment. We thank JC Cahill for comments on a previous version of the manuscript.

### Funding

Support was provided by the US Department of Energy, Office of Environment and Science (Grant #DE-FG02-04ER63892). Additional support was provided by the USDA CREES Ecosystem Studies Program, and the NSF Ecosystem Studies and LTREB programs. The funders had no role in study design, data collection and analysis, decision to publish, or preparation of the manuscript.

### Grant Disclosures

The following grant information was disclosed by the authors:
US Department of Energy.
Office of Environment and Science: #DE-FG02-04ER63892.
USDA CREES Ecosystem Studies Program.

### Competing Interests

The authors declare there are no competing interests.

### Author Contributions

- Ava M. Hoffman analyzed the data, contributed reagents/materials/analysis tools, wrote the paper, prepared figures and/or tables, reviewed drafts of the paper.
- Meghan L. Avolio, Alan K. Knapp and Melinda D. Smith conceived and designed the experiments, performed the experiments, contributed reagents/materials/analysis tools, reviewed drafts of the paper.

### Field Study Permissions

The following information was supplied relating to field study approvals (i.e., approving body and any reference numbers):
    Field experiments were approved by Kansas State University.

### Microarray Data Deposition

The following information was supplied regarding the deposition of microarray data:
    All processed and normalized data:
    Hoffman, Ava (2018): Climate_extremes_tallgrass_prairie_microarray_analytics. figshare. https://doi.org/10.6084/m9.figshare.5627509.v2.
    All raw data:
    Hoffman, Ava; L. Avolio, Meghan; Knapp, Alan K.; D. Smith, Melinda (2018): Climate_extremes_tallgrass_prairie_raw_microarray. figshare.
    https://doi.org/10.6084/m9.figshare.5627425.v2.

## Data Availability

All processed and normalized data and code/scripts:

DOI: 10.6084/m9.figshare.5627509.

## Supplemental Information

Supplemental information for this article can be found online at http://dx.doi.org/10.7717/peerj.4394#supplemental-information.

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
