# Peer review of "Codominant grasses differ in gene expression under experimental climate extremes in native tallgrass prairie"

_PeerJ, doi:10.7717/peerj.4394_

## Round 0.1 · original submission · Minor Revisions

Dear Ava,

Thank you for the submission of the paper on gene expression during climate extremes. The work is both interesting and very valuable. The reviewers have suggested a number of revisions. If you decide to address the changes, please note the figure 2 font sizes and captions, please make these consistent.

With kind regards,

Melanie Zeppel

·

Basic reporting

The article was for the most part easy to understand. There are a few items that could be clarified in order to be more understandable to a broader audience.

Experimental design

As the authors acknowledge, the sample size was a low and microarrays are not as detailed as an RNA seq study, but the study is important and provides an important narrative toward understanding how species may respond to climate change.

I have never used microarrays, so I will defer to another reviewer on those statistics.

Validity of the findings

No comment.

Additional comments

Great article! We're starting an RNA-seq assisted migration study on cool-season grasses in North Dakota, so your article was of particular interest to me.

·

Basic reporting

It is well written, clearly stating the results and the implications for this field of research.
Meets all the guidelines of PeerJ wrt references etc.

Experimental design

Good

Validity of the findings

Good

Additional comments

Review of PeerJ manuscript;
Codominant grasses differ in gene expression under experimental climate extremes in native tallgrass prairie.
Ava M Hoffman, Meghan L Avolio, Alan K Knapp, Melinda D Smith
This is a small but worthwhile analysis of the differences in gene expression between two grass species grown under environmental stress. It is well written, clearly stating the results and the implications for this field of research. I recommend acceptance and publication of this manuscript.
However, I do have a number of minor comments that should be addressed.
• Line 37 – Cite IPCC 2013 but there are no details in the reference list under IPCC. So add Stocker et al 2013 to the text or put the details under IPCC

• Following sentence is incomplete - improving our understanding of what and there is no full stop.
52 "However, most molecular studies of plant responses to drought and heat stress are focused on
53 model organisms with limited ecological relevance (Leakey et al. 2009), although awareness and
54 sequencing costs are improving our understanding (Voesenek et al. 2014; Ellegren 2014) "

• In the site description (line 99) the plot sizes are given as 3 x 6m but in the legend to Online resources Supplementary methods line 3 it states 2 x 6 m;

• line 276 – 'both histone acetyltransferases were upregulated" implies only two exist in the species. Add that the two histone acetyltransferases identified as differentially expressed in A.gerardii were upregulated.

I do have two comments that could be taken into account for future studies;
1. The study is limited in that only two individuals were sampled for each species under each of the treatments but I agree with the authors arguments that this is sufficient for this preliminary study.
2. Hybridisation of cDNA synthesised from leaf RNA against maize ear tissue may have limited the type of genes identified.
However, I also recognise that there is a limit to the resources available for a study such as this and despite the above reservations I think these results add to the information available in this area of research.

---

## Round 0.2 · accepted · Accept

Dear Ava,

Congratulations on your acceptance. May I encourage you to publish more of your excellent work and keep going with your research.